# Electrocatalytic continuous flow chlorinations with iodine(I/III) mediators

Tuhin Patra [1,2], Sagar Arepally [1], Jakob Seitz[1] & Thomas Wirth [1] ✉

Electrochemistry offers tunable, cost effective and environmentally friendly alternatives to carry out redox reactions with electrons as traceless reagents. The use of organoiodine compounds as electrocatalysts is largely under-developed, despite their widespread application as powerful and versatile reagents. Mechanistic data reveal that the hexafluoroisopropanol assisted iodoarene oxidation is followed by a stepwise chloride ligand exchange for the catalytic generation of the dichloroiodoarene mediator. Here, we report an environmentally benign iodine(I/III) electrocatalytic platform for the in situ generation of dichloroiodoarenes for different reactions such as mono- and dichlorinations as well as chlorocyclisations within a continuous flow setup.

Hypervalent iodine reagents are classic compounds in organic synthesis. They are environmentally friendly and non-metallic reagents that have recently become again significant to organic chemists as they can affect many different reactions, for example, oxidative functionalizations or rearrangements. They are mild reagents and can often replace transition-metal catalysed protocols[1–3]. Willgerodt used elemental chlorine in the oxidation of iodoarenes for the synthesis of (dichloroiodo)arenes, the first hypervalent iodine reagents[4]. However, improvement of the stability of this class of reagents through structural analysis is explored continuously to further exploit their reactivities[5]. Evidently, instead of their stoichiometric use, an in situ electrocatalytic generation represents an intriguing solution in terms of atom economy and sustainability[6–8]. In practice, however, sluggish interfacial electron transfer rates from the electrode to the substrate constitute a major drawback in realising the full potential of organic electrosynthesis[9,10]. Indirect electrolysis with mediator molecules has emerged as an important strategy to facilitate electron transfer between the electrode and substrate, which can improve the efficiency of electrosynthetic processes[11].

However, the practical realisation of an electrochemical process poses a significant challenge due to the relatively high oxidation potentials of aryl iodides in comparison to the substrates of interest. So far, aryl iodides are predominantly electrolysed in the absence of substrates and subsequently used as stoichiometric reagents in ex-cell mediation (Fig. 1)[12–17]. Recent elegant work from the Powers and Xu groups unveiled the impending potential of iodoarene electrocatalysis[18,19]. However, the use of constant potential electrolysis

and the obligatory requirement of supporting electrolytes in high concentration are yet to be addressed. This further impairs the process in terms of sustainability as both the mediator and the supporting electrolyte usually become part of the waste stream. Francke and coworkers have addressed this by tethering the supporting electrolyte with the iodoarene mediator to improve the atom economy[15,16,20]. Aiming to minimise the amount of supporting electrolytes along with improved efficiency, our group has developed a flow electrolysis platform for the stoichiometric generation of hypervalent iodine compounds, followed by their in-line use in diverse synthetic applications[21,22]. Though this approach represents an important step forward regarding sustainability, the central issue of in-cell electrocatalytic generation and use of hypervalent iodine compounds remains challenging.

Up to now, manganese has prominently been used as the preferred transition metal-based electrocatalyst for vicinal dichlorination of alkenes[23–25]. In sharp contrast, the development of organoelectrocatalysed chlorinations seems to be limited to a sole example of a selenium-catalysed process[26]. Willgerodt's reagent (dichloroiodobenzene, PhICl₂) has been extensively used as a stoichiometric chlorinating reagent[27–33] after its first isolation in 1886[34]. It has also been used as an oxidant in various processes[35]. Astonishingly, a catalytic method for generating Willgerodt-type reagents using a chemical oxidant for chlorination reactions has only been developed recently[36]. Even some enantioselective attempts using a similar approach have been published[37]. However, the electrocatalytic use of such hypervalent iodine compounds for chlorination reactions is an uncharted area.

[1]School of Chemistry, Cardiff University, Park Place, Main Building, Cardiff, Cymru/Wales, UK. [2]Present address: School of Basic Sciences, Indian Institute of Technology Bhubaneswar, Argul, Odisha, India. ✉e-mail: wirth@cf.ac.uk

**A. Aryl iodide mediated electrosynthesis**

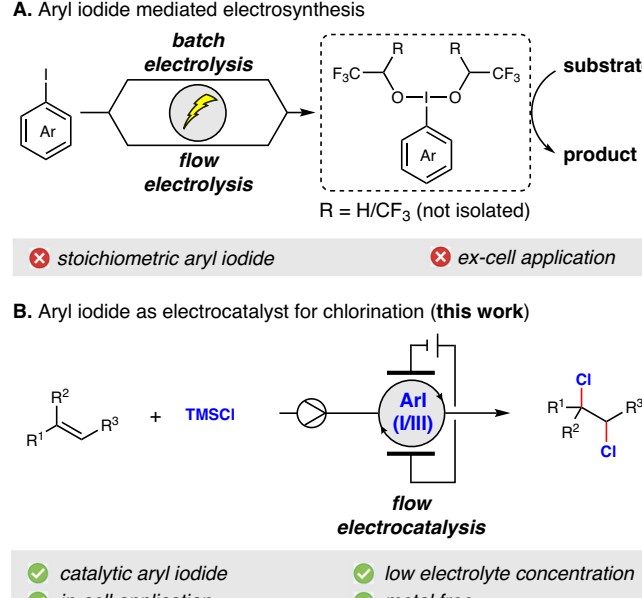

❌ *stoichiometric aryl iodide*     ❌ *ex-cell application*

**B. Aryl iodide as electrocatalyst for chlorination (this work)**

✅ *catalytic aryl iodide*     ✅ *low electrolyte concentration*
✅ *in-cell application*     ✅ *metal free*

**Fig. 1 | Aryl iodides in electrochemistry. A** Aryl iodide mediated electrosynthesis. **B** Aryl iodide as electrocatalyst for chlorination (This work).

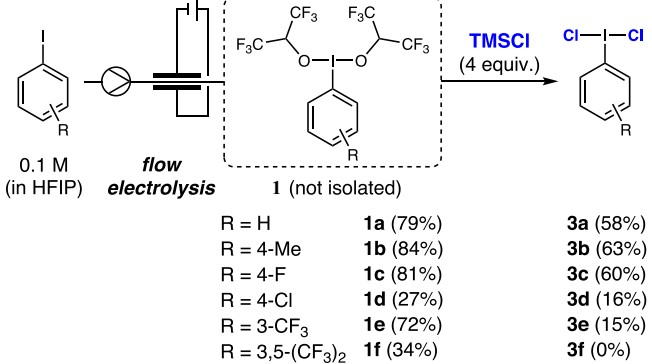

| R = H | **1a** (79%) | **3a** (58%) |
| R = 4-Me | **1b** (84%) | **3b** (63%) |
| R = 4-F | **1c** (81%) | **3c** (60%) |
| R = 4-Cl | **1d** (27%) | **3d** (16%) |
| R = 3-CF$_3$ | **1e** (72%) | **3e** (15%) |
| R = 3,5-(CF$_3$)$_2$ | **1f** (34%) | **3f** (0%) |

**Fig. 2 | (Dichloroiodo)arene synthesis.** Formation of ArICl$_2$ through electrochemical oxidation followed by chloride ligand exchange. (HFIP: 1,1,1,3,3,3-hexafluoroisopropanol).

This is surprising given the fact that the electrochemical synthesis of the corresponding (difluoroiodo)benzene (PhIF$_2$) is known since 1960[38] and has been used in batch[39,40] and flow protocols[22] more recently.

In this work, we build on the advantages of a flow electrochemical approach[41–45] and report herein an organoiodine(I/III) electrocatalytic platform for the generation of (dichloroiodo)arenes as Willgerodt-type reagents for the dichlorination of non-activated alkenes. Furthermore, we extend this approach successfully to a diverse range of chlorination reactions with a detailed mechanistic understanding.

## Results

Based on our previous experiences on the electrochemical generation of hypervalent iodine compounds[20], our initial efforts were directed to find a suitable chloride source for the generation of (dichloroiodo) arenes by ligand exchange of electrochemically generated hypervalent iodine reagents ligated with fluorinated alcohols (**1**). Though such reagents are not bench stable, they are stable in solution and can be easily synthesized by electrochemical oxidation reactions[11–14,46]. Evaluation of different chloride sources with the electrochemically

synthesized **1a** revealed that such ligand exchange is feasible, see Supplementary Table 1. Notably, chlorotrimethylsilane (TMSCl) was found to be optimal for a fast chloride ligand exchange. The high oxophilicity of the silicon atom might be beneficial in the ligand exchange involving hexafluoroisopropoxide as the leaving group. As shown in Fig. 2, different iodoarenes could be electrochemically oxidised, followed by a facile chloride ligand exchange with TMSCl to generate corresponding (dichloroiodo)arenes **3**. Only strong electron-deficient arenes produce low yields of the corresponding (dichloroiodo)arenes **3**.

However, in contrast to the electrochemical oxidation step, the efficacy of the ligand / chloride exchange is restricted by the competing oxidative side reactions resulting in the regeneration of the aryl iodide[19]. Furthermore, purification of these (dichloroiodo)arenes is problematic owing to their low stability and vulnerability to light and low pressure[47]. We envisioned that such a scenario could set an ideal platform for the development of an electrocatalytic method for the in-cell generation of (dichloroiodo)arenes for subsequent applications in chlorination reactions.

To explore this possibility, initially, the electrochemically generated **3a** was applied in a stoichiometric dichlorination reaction with alkene **4** in an ex-cell fashion. Indeed, the formation of the desired product **5** confirmed that the development of an electrocatalytic version could be feasible. Encouraged by this, we began our investigations by varying different parameters for electrocatalytic dichlorination reactions. Judiciously, 4-iodotoluene was selected as a preferred electrocatalyst due to its low oxidation potential combined with excellent performance for the electrochemical generation of hypervalent iodine compound followed by efficient chloride ligand exchange under the reaction conditions (Fig. 2). We used a commercially available electrochemical microflow reactor for rapid screening of different reaction parameters[48]. Electrochemical microflow reactors are a useful tool for performing electrochemical transformations in a more efficient way owing to small inter-electrode gap, large electrode surface-to-volume ratio, high mass transfer and intensified reaction conditions[49,50,51,52].

Extensive screening of different reaction parameters revealed that a combination of 25 mol% of 4-iodotoluene and 5 mol% of a supporting electrolyte (tetramethylammonium hexafluorophosphate, Me$_4$NPF$_6$) furnished the desired dichlorinated product **5** in just 12 min (Table 1, entry 1). Importantly, expensive electrode combinations could be replaced by more practical and cheap graphite electrodes without affecting the reaction outcome (entry 2). Control experiments confirmed that both the electrocatalyst and the electricity were indispensable for the formation of the desired product in high yields (entries 3 and 4). Similarly, a catalytic amount of supporting electrolyte was also beneficial for the reaction (entry 5). Generally, metal chloride salts were not as effective as TMSCl due to their poor solubility leading to the blockage of the channel over prolonged runs (entry 6). The use of a batch electrochemical cell for a similar transformation, even with higher loading of supporting electrolyte, was less efficient compared to the electrochemical microflow reactor (entry 7).

With the optimised reactions conditions in hand, we explored the substrate generality for the electrocatalytic dichlorination. Firstly, different non-activated terminal alkenes were investigated and all of them delivered the desired dichlorinated products in moderate to good yields (Fig. 3, compounds **5**–**21**). Simple long chain alkenes were engaged easily (**5**–**7**). Different oxidation and reduction-sensitive functional groups including esters (**8**–**10**), nitriles (**9**), alcohols (**11**), free carboxylic acids (**12**), keto groups (**23**), fluoro- (**24**), chloro- (**28**), bromo- (**13**) and iodo substituents (**14**), sulphones (**16**) and phosphonate esters (**17**) were well tolerated. Even a free carboxylic acid was amenable to the reaction conditions without any decarboxylation due to the strong hydrogen bonding with the 1,1,1,3,3,3-hexafluoroisopropanol (HFIP)

**Table 1 | Optimisation for electrocatalysed dichlorination reaction of alkenes**

Ph—CH₂CH₂CH=CH₂ (**4**) + TMSCl (4 equiv.) → 4-iodotoluene (25 mol%), Me₄NPF₆ (5 mol%), HFIP/MeCN (2:1, 0.1 M), GC(+)/Pt(−), 2 mA/cm² (3 F), rt, t_R = 12 min → Ph—CH₂CH₂CHCl—CH₂Cl (**5**)

| entry | deviation from the above conditions | yield of 5 (%)[a] |
|---|---|---|
| 1 | none | 78 |
| 2 | Gr(+)/Gr(−) | 82 (80)[b] |
| 3 | no 4-iodotoluene | 9 |
| 4 | no electricity | 0 |
| 5 | no electrolyte | 44 |
| 6 | CsCl instead of TMSCl | 33 |
| 7 | batch electrolysis[c] | 61 |

[a] Yields were determined by gas chromatography using n-dodecane as an internal standard.
[b] Isolated yield of **5**.
[c] 50 mol% of supporting electrolyte Me₄NPF₆ was used.

solvent (**12**)[17]. Expectedly, the alkene tethered with a distal aryl iodide delivered the desired product, even in the absence of the electrocatalyst (**14**). This further highlights the crucial role of an aryl iodide mediator in our reaction conditions. Furthermore, an internal alkyne was tolerated in the presence of a reactive terminal alkene to deliver **15** in an acceptable yield. Different nitrogen and oxygen-based heterocycles were successfully incorporated without any difficulties (**18–20**). A (+)-Fenchol derivative returned product **21** in a 1:1 diastereomeric ratio.

Different internal alkenes were investigated in electrocatalytic dichlorination conditions. Methyl cinnamate was converted to the corresponding dichloro derivative **22** with exclusive anti-diastereoselectivity. Similarly, different chalcones were successfully dichlorinated in a highly diastereoselective fashion (**23, 24**). A systematic extension of the scope towards non-activated internal alkenes provided an uneven mixture of diastereoisomers[53]. Consistent with the reactivities of **3a**, while (E)-alkenes led to the anti-configured products **26** and **28** as major products, (Z)-alkenes favoured the syn-adducts (**27** and **29**)[54]. Remotely substituted internal alkenes were, however, found to be unreactive. Notably, sterically demanding trisubstituted alkenes also participated in vicinal dichlorination reactions (**30, 31**). Interestingly, 1,1,2-triphenylethylene delivered the selective monochlorination product **32**. The initially formed triphenyldichloroethane is known to undergo facile hydrogen chloride elimination in alcoholic solutions to produce **32** quantitatively[55]. The rate of uncatalysed background reactions was found to be larger for these highly electron-rich trisubstituted alkenes.

Taking advantage of the reactivities of **3a**, different unsaturated carboxylic acids were successfully employed in aryl iodide electrocatalysed chlorocyclisations leading to products **33** and **34**. Finally, the concept was also successfully extended to develop an aryl iodide electro-catalysed monochlorination process. A series of 1,3-dicarbonyl as well as β-keto ester derivatives were employed effectively irrespective of substitution patterns providing compounds **35–39** in very good yields.

After developing versatile methods for aryl iodide electrocatalysed chlorination reactions, subsequent efforts were directed towards corroborating the postulated mechanism involving ArICl₂ (**3**) as the active mediator. After confirming the formation of **3** from stoichiometric experiments (Fig. 2), we independently synthesised **3a** and **3b** to compare their reactivities in alkene dichlorination reactions. The formation of PhICl₂ (**3a**) from PhI[OCH(CF₃)₂]₂ (**1a**) was systematically investigated by ¹H and ¹⁹F NMR spectroscopy to identify the active chlorinating intermediate between **2a** and **3a**. After identifying the characteristic ¹H NMR signals for **1a** and **3a**, the stoichiometry of TMSCl was varied with respect to **1a**. Intriguingly, the unsymmetrical hypervalent iodine intermediate **2a** was successfully identified (Fig. 4a)[56]. The different intermediates and the presence of **2a** were further confirmed by ¹⁹F NMR analysis (Supplementary Figure S4). The formation of **3a** from **1a** follows a stepwise pathway involving **2a** as an intermediate. Intermediate **2a** was instantaneously converted to **3a** in an exclusive fashion in the presence of more than two equivalents of TMSCl, along with the partial regeneration of iodobenzene. Notably, all our attempts to generate **2a** from **3a** were unsuccessful. This further indicates that **3a** is likely the active intermediate for chlorination reactions under the optimised reaction conditions.

Consequently, the proficiency of dichloro(p-tolyl)-λ³-iodane (**3b**) was investigated for the dichlorination of alkenes. The unproductive control experiments involving **3b** in the absence of HFIP highlight the indispensable role of HFIP in the chlorination steps. Expectedly, **8** could be effectively generated using a stoichiometric amount of **3b** (Fig. 4b). Clearly, **3b** was also effective in carrying out the catalytic cycle successfully (Fig. 4c). As a control, direct electrolysis of a TMSCl solution in the absence of 4-iodotoluene, followed by the addition of the alkene in an "ex-cell" fashion, did not result in any dichlorinated product formation. All these experiments exclude the involvement of elemental chlorine and undoubtedly confirm the catalytic role of **3b** in the C–Cl bond-forming steps. The C–Cl bond forming step for dichlorination could be of partially radical nature as the addition of BHT to the reaction with stoichiometric **3b** reduced the NMR yield to 13% (Fig. 4d). The addition of TEMPO, however, did not affect the yield, in this reaction, a TEMPO adduct to the alkene was identified by mass spectrometry. Such partial radical nature for dichlorination reactions could also be converged from the fact that both (E)- and (Z)-alkenes returned a mixture of diastereomers (Fig. 3, compounds **26–29**). Notably, **3a** is also known to act in radical chlorination processes[57].

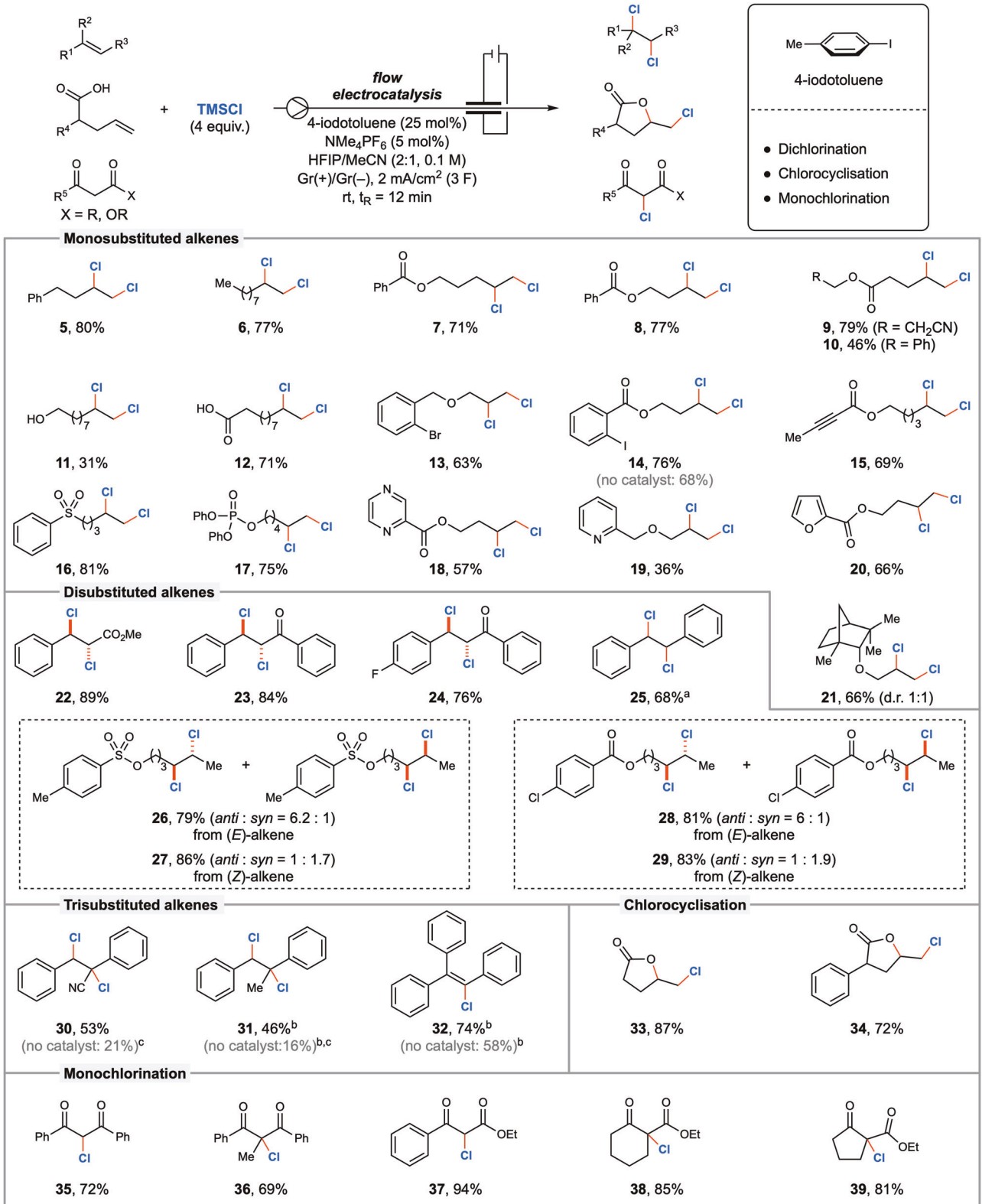

**Fig. 3 | Scope of the electrocatalysed chlorination reaction.** Reaction conditions: alkene (0.5 mmol), TMSCl (2.0 mmol), 4-iodotoluene (25 mol %), NMe$_4$PF$_6$ (5 mol%) in HFIP/MeCN (2:1 v/v, 0.1 M), graphite electrodes, electrode distance 0.5 mm, 2 mA/cm$^2$ (3 F), 12 min residence time at room temperature, 600 μL reactor volume, flow rate 0.05 mL/min, isolated yields. **a** HFIP/MeCN/CH$_2$Cl$_2$ (4.2:3.1:2 v/v, 0.1 M). **b** HFIP/MeCN/CH$_2$Cl$_2$ (5.2:3.1:1 v/v, 0.1 M). **c** Yields were determined by $^1$H NMR analysis of the crude reaction mixture.

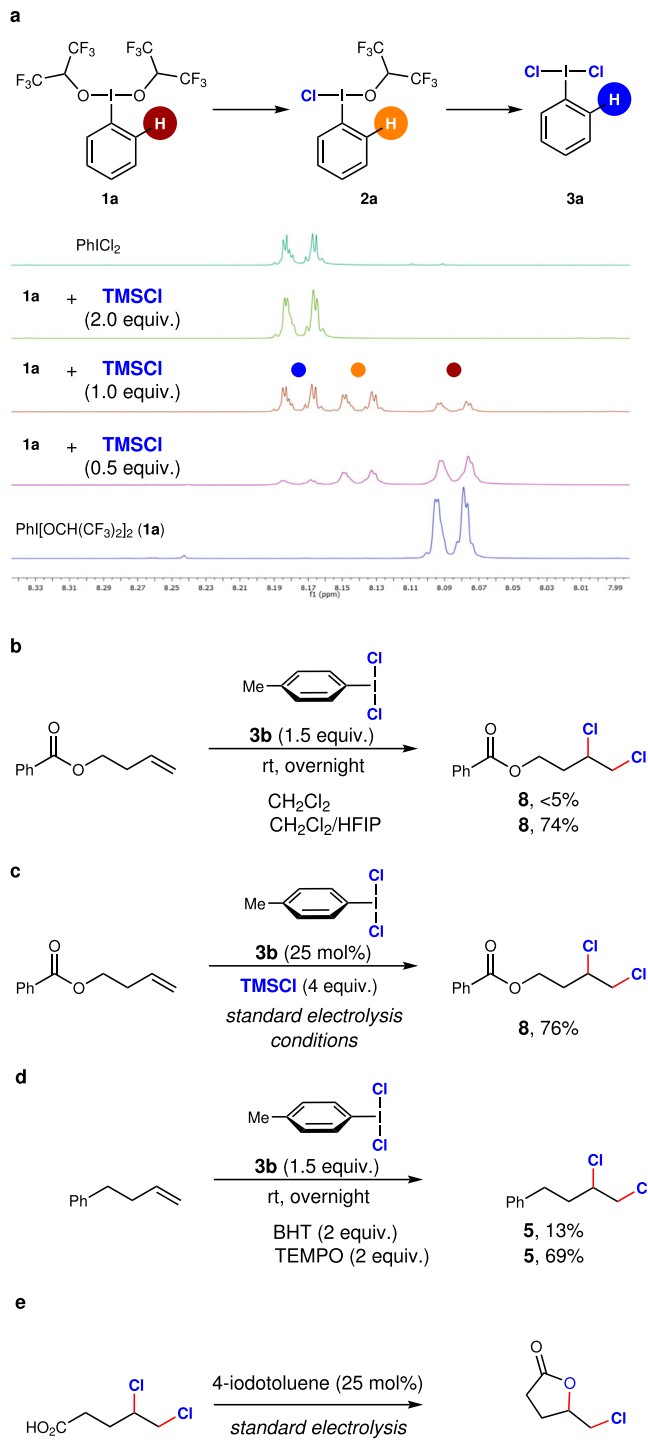

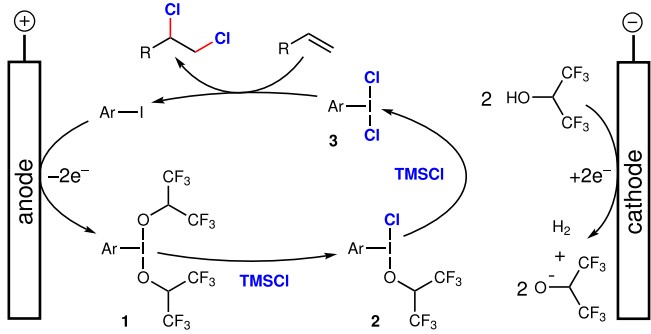

**Fig. 5 | Proposed mechanism.**

in stabilizing electrochemically generated iodine(III) compounds[7,11,20,45]. Thirdly, the low p$K_a$ (9.4) of HFIP allows clean proton reduction as the cathodic half-reaction rendering only $H_2$ as the byproduct of this electrochemical process. Finally, the non-nucleophilic and polar HFIP has a positive impact on the rate and selectivity of transformations involving radical and ionic intermediates and hence is often used in iodine(III)-mediated reactions[40,58].

Combining all these facts together, the proposed mechanism begins with the two-electron anodic oxidation of aryl iodide to form **1**. The two-electron reduction of two HFIP solvent molecules at the cathode readily generates the 1,1,1,3,3,3-hexafluoroisoprop-oxide ligands for **1** with the concomitant liberation of hydrogen gas (Fig. 5). Next, **1** undergoes stepwise ligand exchange with TMSCl to generate **3** as the active species. Finally, **3** participates in the C–Cl bond forming reactions with the alkene to produce the chlorinated product and regenerates the electrocatalyst. The C–Cl bond forming steps involving hypervalent iodine compounds are mechanistically consistent with previous reports[53,59]. Significantly, the poor nucleophilicity of the 1,1,1,3,3,3-hexa-fluoroisopropoxide ensures a very high level of selectivity for chlorination reactions.

Finally, with a view on the enantioselective version of these dichlorinations, initial studies were performed with chiral iodoarene catalysts. Preliminary validation for chirality induction showed that dichloride product **20** was obtained with a non-significant enantio-meric ratio of 54:46 under identical flow electrochemical condi-tions (Fig. 6).

## Discussion

4-Iodotoluene was successfully introduced as an efficient electro-catalyst for dichlorinations, chlorocyclisations and monochlorinations by exploiting the reactivities of Willgerodt-type reagents. Mechanistic investigations revealed that a stepwise chloride ligand exchange pathway is viable for the generation of the active symmetrical hyper-valent iodine intermediate. With the wide-ranging applications of hypervalent iodine reagents already known in organic synthesis, we anticipate iodine(I/III) electrocatalysis will gain comprehensive interest in the coming years.

## Methods

### General procedure for the dichlorination of alkenes (using 5 as an example)

4-Iodotoluene (55 mg, 0.25 mmol, 0.25 equiv.) and Me₄NPF₆ (11 mg, 0.05 mmol, 0.05 equiv.) were added to an oven-dried vial equipped with a Teflon-coated magnetic stir bar. Dry MeCN (3.1 mL), but-3-en-1-ylbenzene (132 mg, 1.0 mmol, 1.0 equiv.) and TMSCl (507 μL, 4.0 mmol, 4.0 equiv.) were added sequentially and the reaction mixture was pre-stirred until a clear solution was obtained. Dry HFIP (~6.2 mL) was added to the reaction mixture to reach 10 mL volume (0.1 M). The mixture was stirred until homogeneous and placed in a

**Fig. 4 | Mechanistic investigations. a** Stepwise chloride exchange (¹H NMR, 500 MHz, CDCl₃). **b** Stoichiometric competency. **c** Catalytic competency. **d** Involvement of radicals in alkene dichlorination (BHT: butylhydroxytoluene, TEMPO: 2,2,6,6-tetramethylpiperidinyloxy). **e** Control experiment for the possible chlorocyclisation of a γ,δ-dichlorinated acid.

A dichlorination followed by intramolecular substitution can be ruled out for chlorocyclisation reactions as the γ,δ-dichlorinated acid failed to provide **33** under the standard electrochemical reaction conditions (Fig. 4e).

HFIP plays multiple roles in this reaction. Firstly, HFIP has excellent anodic stability and high conductivity suitable for electrochemical oxidation of aryl iodide. Secondly, HFIP is known to play an active role

**Fig. 6 | Preliminary studies for enantioselective variant.** (Mes: 2,4,6-trimethylphenyl).

12 mL disposable syringe. The solution was pumped through the electrochemical setup with a fixed flow rate of 0.05 mL/min to give a residence time of 12 min in the active part of the reactor, equipped with graphite electrodes separated by 0.5 mm FEP spacer. The reaction mixture was subjected to a constant current electrolysis by applying 24 mA current (current density of 2 mA/cm$^2$ with electrode surface area of 12 cm$^2$). This flow rate and concentration delivered 3 F per mole of charge to the reaction mixture. The first 1.5 reactor volumes (0.9 mL) were discarded to reach an equilibrium. After which, the reaction output was collected in a vial for 100 min (5 mL) and then the reaction was stopped after the collection vial was removed. The power supply was turned off and the reactor was washed by passing MeOH and acetone. A 5 μL aliquot from this crude reaction mixture was analyzed by GC/MS. The rest of the reaction mixture was concentrated, and the residue was purified by flash column chromatography to afford the desired product **5** as a colorless oil (81 mg, 80%).

## Data availability

Detailed experimental procedures and characterization data are provided in the Supplementary Information. Information about the data that underpins the results presented in this article can be found in the Cardiff University data catalog at https://doi.org/10.17035/d.2024.0328125051. All data are available from the corresponding author upon request.

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

## Acknowledgements

Generous financial support from the Engineering and Physical Sciences Research Council (EPSRC, grant EP/T019719/1, T. P. and S. A.) and the School of Chemistry, Cardiff University (PhD studentship to J. S.) are gratefully acknowledged. The authors thank Dr. Bethan Winterson and Dr. Nasser Amri for experimental assistance and helpful discussions and Dr. Nisar Ahmed for proofreading an early version of the manuscript.

## Author contributions

T. W. conceptualized the work, T. P. developed the methodology, and T. P., J. S., and S. A. performed all the experiments and analyzed the data. T. P., S. A., and T. W. prepared the manuscript.

## Competing interests

The authors declare no competing interests.
