## [Peer Review File · Nature Communications]

Electrocatalytic Continuous Flow Chlorinations with Iodine(I/III) MediatorsREVIEWER COMMENTS

Reviewer #1 (Remarks to the Author):

In this manuscript, Wirth and coauthors report an environmentally benign iodine(I/III) electrocatalytic platform for the in-situ generation of dichloroiodoarenes (ArICl_2), which was suitable for various reactions such as mono- and dichlorinations as well as chlorocyclizations within a continuous flow setup. This strategy demonstrates that mechanistic data reveal the hexafluoroisopropanol assisted iodoarene oxidation is followed by a stepwise chloride ligand exchange for the catalytic generation of the ArICl_2 mediator. However, the authors do not sufficiently investigate the specific process of chlorination of the substrates, and the mechanistic study needs to be improved. Furthermore, the authors have not conducted further research on the applicability of this method, such as the derivatization of the products. From conceptual aspect, there are several works have been published (see *J. Am. Chem. Soc.* 2017, 139, 15548–15553; *Acc. Chem. Res.* 2020, 53, 3, 547–560). Overall, I think this work is not suitable to be published in *Nature Communications*. To improve the quality of this manuscript, there are some comments and suggestions should be addressed:

1. In Table 2, concerning the carboxylic acid cyclization reactions, is the mechanism similar to the dichlorination of olefins? Is it possible that a dichlorinated intermediate is first formed, followed by an intramolecular nucleophilic substitution to yield the product, or does a monochlorinated intermediate first form, which then undergoes oxidation to generate a carbocation that subsequently undergoes intramolecular cyclization to form the product? Additionally, the monochlorination of 1,3-dicarbonyl compounds: is this a radical-type reaction, or an ionic-type reaction?
2. Whether the intermediate 2a has been identified?
3. In the introduction of the manuscript, the novelty of the experimental method is confusing and requires further emphasis on the characteristics of the reaction.
4. Why hexafluoroisopropanol plays a significant role in the chlorination process is not further explained in the mechanistic study.
5. The experimental study in Figure 3d does not provide any supportive evidence for the mechanistic study.
6. Was the potential product (3,4-bis((1,1,1,3,3,3-hexafluoropropan-2-yl)oxy)butyl)benzene detected, or the possible product (4-chloro-3-((1,1,1,3,3,3-hexafluoropropan-2-

yl)oxy)butyl)benzene?

7. Some related reference should be properly cited: DOI: 10.1093/nsr/nwad187;
10.1038/s41467-019-10928-0.

Reviewer #2 (Remarks to the Author):

This is a good paper and can be published in Nature Communications. While flow electrochemistry in concert with catalytic hypervalent iodine is known (indeed developed by these authors), the lack of chlorination methods using flow Echem makes this report an important advance. That the technique works is not particularly surprising, all the individual components have been known for some time, but it is good to see ArICl_2 chlorination and flow electrochemical catalysis brought together. TMS-Cl is an attractive chloride source, as this is often waste for other chemical reactions and a low-energy source.

A particular strength of this paper is the supporting information. It is sublime. All the details are given, including photos of the set-up. Even more impressive is that the authors have presented imperfect spectra and simply pointed out cases where overlapping peaks etc prevent firm assignments - we need to see more of this in the literature, rather than feeling pressure to always produce special perfect spectra for the SI.

Reviewer #3 (Remarks to the Author):

This manuscript by Wirth and co-workers describes a very attractive method to enable the mono-, di- and chlorocyclisation of an array of simple, unfunctionalized substrates through an electrocatalytic flow platform. The reaction relies on an I(I)/I(III) cycle and utilises inexpensive 4-iodotoluene as the catalyst (25 mol%) with TMS-Cl (4 eq.) serving as the chloride source. In situ generation of pTolICl_2 is achieved electrochemically (graphite electrodes) with HFIP assisting the oxidation and being replaced by Cl through ligand exchange at the iodine centre. The introduction of the paper is compelling and I commend the authors on highlighting the pioneering work of Willgerodt, after whom the parent reagent (PhICl_2) is named. The authors may also wish to cite recent structural work on this reagent (Synthesis 2019, 51, 4408–4416). Seminal contributions from the labs of Power and

Xu are also prominently cited and I find this both refreshing and scholarly. As shown in Figure 1, the key advances of this chemistry include the in-cell application (as opposed to ex-cell approaches for the preparation of I(III) intermediates), the low electrolyte concentration, metal-free conditions and comparatively low catalyst loadings of pToll. Collectively, these are significant advances in a very active field of contemporary research. The selection of catalyst and chloride source as convincingly demonstrated in Figure 2 and the advantages of TMSCl in terms of facile ligand exchange to generate the ArICl₂ product far outweigh any concerns regarding atom economy. Similarly, the choice of graphite electrodes is demonstrated in the vicinal dichlorination of alkene 4 (to form 5 in 80% yield) and the scope in Table 2 is very convincing. Examples of 1,2-dichlorination (with both internal and terminal alkenes), chlorolactonisation and the alpha chlorination of 1,3-dicarbonyl compounds has been validated. The postulated mechanism is supported by mechanistic work and this includes (1) a demonstration of stepwise chloride exchange, (2) a stoichiometric comparison (with and without the HFIP) and the addition of BHT and TEMPO as radical traps. My only slight concern is the potential for uncatalyzed background reactions with highly electron-rich alkenes (e.g. 29, 30 and 31), as this is known with many halofunctionalisation reactions using stoichiometric oxidants. It would be helpful to run controls in these cases. Since the very moderate enantioselectivity has been reported in the vicinal dichlorination of alkenes by Gilmour (cited as reference 37), I am curious to know if the authors tried a chiral ArI catalyst in any of the transformations reported? Can the authors comment a little more on the role of the HFIP in the oxidation step? This is interesting and potentially expansive. Overall, I really enjoyed reading this paper and I recommend publication of the work in Nature Communications. The chemistry is clearly powerful, well-demonstrated and residence times of 12 minutes far outcompetes conventional approaches!

Revision of manuscript id NCOMMS-24-17595

(Electrocatalytic Continuous Flow Chlorinations with Iodine(I/III) Mediators)

We are very thankful for the positive comments and suggestions of all reviewers. Our detailed feedback and comments are below.

【Reviewer 1】

Statements:

In this manuscript, Wirth and coauthors report an environmentally benign iodine(I/III) electrocatalytic platform for the in-situ generation of dichloriodoarenes (ArICl_2), which was suitable for various reactions such as mono- and dichlorinations as well as chlorocyclizations within a continuous flow setup. This strategy demonstrates that mechanistic data reveal the hexafluoroisopropanol assisted iodoarene oxidation is followed by a stepwise chloride ligand exchange for the catalytic generation of the ArICl_2 mediator. However, the authors do not sufficiently investigate the specific process of chlorination of the substrates, and the mechanistic study needs to be improved. Furthermore, the authors have not conducted further research on the applicability of this method, such as the derivatization of the products. From conceptual aspect, there are several works have been published (see J. Am. Chem. Soc. 2017, 139, 15548–15553; Acc. Chem. Res. 2020, 53, 3, 547–560). Overall, I think this work is not suitable to be published in Nature Communications. To improve the quality of this manuscript, there are some comments and suggestions should be addressed:

Response to the statements of Reviewer 1:

We thank the reviewer for providing comments and suggestions to improve the quality of the manuscript. The first reference above was already included in our manuscript (ref. 23), the second one is now also included as reference 24.

Comment 1-1: “In Table 2, concerning the carboxylic acid cyclization reactions, is the mechanism similar to the dichlorination of olefins? Is it possible that a dichlorinated intermediate is first formed, followed by an intramolecular nucleophilic substitution to yield the product, or does a monochlorinated intermediate first form, which then undergoes oxidation to generate a carbocation that subsequently undergoes intramolecular cyclization to form the product? Additionally, the monochlorination of 1,3-dicarbonyl compounds: is this a radical-type reaction, or an ionic-type reaction?”

Response: In response to the reviewer’s question, we have verified the involvement of a dichlorinated intermediate in the chlorocyclization reaction through independent synthesis

of the starting material. The result of this mechanistic experiment is now included in Figure 3e. The following statement is now included in the manuscript (page 5):

A dichlorination followed by intramolecular substitution can be ruled out for chlorocyclisation reactions as the γ,δ -dichlorinated acid failed to provide **33** under the standard electrochemical reaction conditions (Fig. 3e).

Additionally, we investigated the involvement of radicals in the monochlorination and found the results to be similar to those to the dichlorination reactions. This data is now included in the supplementary information (Page 13, Figure 6). Also, we have included further points and references to clarify the mechanism in detail in the revised manuscript (page 5):

Such partial radical nature for dichlorination reactions could also be converged from the fact that both *E*- and *Z*-alkenes returned mixture of diastereomers (Table 2, entries **26-29**). Notably, **3a** is also known to act in radical chlorination processes⁵⁷.

Comment 1-2: “Whether the intermediate **2a** has been identified?”

Response: Intermediate **2a** was identified through spectroscopic analysis, as already detailed in the original manuscript on page 4, after ((Figure 3)):

Intriguingly, the unsymmetrical hypervalent iodine intermediate **2a** was successfully identified (Fig. 3a)⁵⁶.

The relevant spectroscopic data can be found in the supplementary information, Figures 3 and 4.

Comment 1-3: “In the introduction of the manuscript, the novelty of the experimental method is confusing and requires further emphasis on the characteristics of the reaction.”

Response: In response to the reviewer’s comment, we emphasize that the in-cell flow electrocatalytic generation and use of hypervalent iodine compound for chlorination reactions are novel aspects of this work. We hope that we have highlighted this unambiguously in the introduction. Only considering the chlorination methodology by overlooking the key iodine (I/III) electrocatalysis part is only a partial evaluation of this work.

Comment 1-4: “Why hexafluoroisopropanol plays a significant role in the chlorination process is not further explained in the mechanistic study.”

Response: We have now included a more detailed discussion on the crucial role of HFIP in the mechanistic section (page 5):

HFIP plays multiple roles in this reaction. Firstly, HFIP has excellent anodic stability and high conductivity suitable for electrochemical oxidation of aryl iodide. Secondly, HFIP is known to play an active role in stabilizing electrochemically generated iodine(III) compounds^{7,11,20,45}. Thirdly, the low pK_a (9.4) of HFIP allows clean proton reduction as the cathodic half-reaction rendering only H_2 as the byproduct of this electrochemical process. Finally, the non-nucleophilic and polar HFIP has a positive impact on the rate and selectivity of transformations involving radical and ionic intermediates and hence is often used in iodine(III)-mediated reactions^{40,58}.

Comment 1-5: “The experimental study in Figure 3d does not provide any supportive evidence for the mechanistic study.”

Response: Figure 3d only indicates that a parallel radical pathway could be operative to the ionic one which is further supported by the diastereomeric ratio obtained for unsymmetrical alkenes (such as products **26** and **27**) as well as by previous reported literature. We have added this discussion and references on page 5:

Such partial radical nature for dichlorination reactions could also be converged from the fact that both *E*- and *Z*-alkenes returned mixture of diastereomers (Table 2, entries **26-29**). Notably, **3a** is also known to act in radical chlorination processes⁵⁷.

Comment 1-6: “Was the potential product (3,4-bis((1,1,1,3,3,3-hexafluoropropan-2-yl)oxy)butyl)benzene detected, or the possible product (4-chloro-3-((1,1,1,3,3,3-hexafluoropropan-2-yl)oxy)butyl)benzene?”

Response: It seems that the poor nucleophilicity of HFIP is inhibiting the formation of C–O bonds in favour of C–Cl bonds in these cases. This clarification has been added on page 5:

The C–Cl bond forming steps involving hypervalent iodine compounds are mechanistically consistent with previous reports^{53,59}. Significantly, the poor nucleophilicity of the 1,1,1,3,3,3-hexafluoroisopropoxide ensures a very high level of selectivity for chlorination reactions.

Comment 1-7: “Some related reference should be properly cited: DOI: 10.1093/nsr/nwad187; 10.1038/s41467-019-10928-0.”

Response: We thank the referee for bringing these papers to our attention, we have added these valuable references as 44 and 45 in the revised manuscript.

【Reviewer 2】

Statements:

This is a good paper and can be published in Nature Communications. While flow electrochemistry in concert with catalytic hypervalent iodine is known (indeed developed by these authors), the lack of chlorination methods using flow Echem makes this report an important advance. That the technique works is not particularly surprising, all the individual components have been known for some time, but it is good to see ArICl_2 chlorination and flow electrochemical catalysis brought together. TMS-Cl is an attractive chloride source, as this is often waste for other chemical reactions and a low-energy source. A particular strength of this paper is the supporting information. It is sublime. All the details are given, including photos of the set-up. Even more impressive is that the authors have presented imperfect spectra and simply pointed out cases where overlapping peaks etc prevent firm assignments - we need to see more of this in the literature, rather than feeling pressure to always produce special perfect spectra for the SI.

Response to the statements of Reviewer 2:

We greatly appreciate the high evaluation of our work and positive feedback. This reviewer did not raise any point to address and is in favour of accepting this work in its current form.

【Reviewer 3】

Statements:

This manuscript by Wirth and co-workers describes a very attractive method to enable the mono-, di- and chlorocyclisation of an array of simple, unfunctionalized substrates through an electrocatalytic flow platform. The reaction relies on an I(I)/I(III) cycle and utilises inexpensive 4-iodotoluene as the catalyst (25 mol%) with TMS-Cl (4 eq.) serving as the chloride source. In situ generation of pTolICl_2 is achieved electrochemically (graphite electrodes) with HFIP assisting the oxidation and being replaced by Cl through ligand exchange at the iodine centre. The introduction of the paper is compelling and I commend the authors on highlighting the pioneering work of Willgerodt, after whom the parent reagent (PhICl_2) is named. The authors may also wish to cite recent structural work on this reagent (Synthesis 2019, 51, 4408–4416). Seminal contributions from the labs of Power and Xu are also prominently cited and I find this both refreshing and

scholarly. As shown in Figure 1, the key advances of this chemistry include the in-cell application (as opposed to ex-cell approaches for the preparation of I(III) intermediates), the low electrolyte concentration, metal-free conditions and comparatively low catalyst loadings of pToll. Collectively, these are significant advances in a very active field of contemporary research. The selection of catalyst and chloride source as convincingly demonstrated in Figure 2 and the advantages of TMSCl in terms of facile ligand exchange to generate the ArICl₂ product far outweigh any concerns regarding atom economy. Similarly, the choice of graphite electrodes is demonstrated in the vicinal dichlorination of alkene 4 (to form 5 in 80% yield) and the scope in Table 2 is very convincing. Examples of 1,2-dichlorination (with both internal and terminal alkenes), chlorolactonisation and the alpha chlorination of 1,3-dicarbonyl compounds has been validated. The postulated mechanism is supported by mechanistic work and this includes (1) a demonstration of stepwise chloride exchange), (2) a stoichiometric comparison (with and without the HFIP) and the addition of BHT and TEMPO as radical traps.

Response to the statements of Reviewer 3:

We appreciate this reviewer's recommendation of our research. This reviewer's suggestions were extremely encouraging, and they indeed helped us to improve our manuscript. A point-by-point response to this reviewer's suggestion/comments is given below.

Comment 3-1: "The authors may also wish to cite recent structural work on this reagent (Synthesis 2019, 51, 4408–4416)."

Response: We have now cited this work as reference 5, with the relevant discussion included on page 1, first paragraph.

However, improvement of the stability of this class of reagents through structural analysis is explored continuously to further exploit their reactivities⁵.

Comment 3-2: "My only slight concern is the potential for uncatalyzed background reactions with highly electron-rich alkenes (e.g. 29, 30 and 31), as this is known with many halofunctionalisation reactions using stoichiometric oxidants. It would be helpful to run controls in these cases."

Response: We thank the reviewer for pointing this out. We have carried out the corresponding control reactions for the highly electron rich alkenes as suggested by the reviewer. Indeed, a larger rate of uncatalysed background reactions were observed. However, the differences in yields between catalysed and uncatalysed reactions are

significant. These results are now included in Table 2 as well as in the discussion (page 4) in the revised manuscript and the relevant spectroscopic data and details can be found in the supplementary information, pages 13–15.

The rate of uncatalysed background reactions were found to be larger for these highly electron-rich trisubstituted alkenes.

Comment 3-3: “Since the very moderate enantioselectivity has been reported in the vicinal dichlorination of alkenes by Gilmour (cited as reference 37), I am curious to know if the authors tried a chiral Arl catalyst in any of the transformations reported?”

Response: We have now explored chiral catalysts and the preliminary validation for chirality induction showed that dichloride product **19** was obtained with an enantiomeric ratio of 54 : 46 under identical flow electrochemical conditions. These results are included as Fig. 5 in the revised manuscript, along with a discussion on page 5. The associated HPLC data and details are provided in the supplementary information, pages 43–45.

Finally, with a view on the enantioselective version of these dichlorinations, initial studies were performed with chiral iodoarene catalysts. Preliminary validation for chirality induction showed that dichloride product **20** was obtained with a non-significant enantiomeric ratio of 54:46 under identical flow electrochemical conditions (Fig. 5).

Comment 3-4: “Can the authors comment a little more on the role of the HFIP in the oxidation step?”

Response: This point has already been addressed in the response to comment 1-4 of reviewer 1, see above.

REVIEWERS' COMMENTS

Reviewer #1 (Remarks to the Author):

The questions aroused from this Reviewer are fully addressed by these authors. I think that the manuscript in present state is appropriate for publishing in Nature Communications. The publication of this interesting work in Nature Communications is recommended.

Reviewer #3 (Remarks to the Author):

I am satisfied that all of the comments and suggestions in my initial report have been fully addressed. Addition experimental work has been performed to answer several open questions and I appreciate the very courteous responses. I enthusiastically endorse acceptance of this manuscript.